# Maintenance of delay-period activity in working memory task is modulated by local network structure

**Dong Yu**[1,2], **Tianyu Li**[1,2], **Qianming Ding**[1,2], **Yong Wu**[1,2], **Ziying Fu**[1,3], **Xuan Zhan**[1,2], **Lijian Yang**[1,2], **Ya Jia**[1,2]*

**1** Institute of Biophysics, Central China Normal University, Wuhan, China, **2** College of Physical Science and Technology, Central China Normal University, Wuhan, China, **3** School of Life Sciences, Central China Normal University, Wuhan, China

* jiay@ccnu.edu.cn

**Data Availability Statement:** All data generated or analysed during this study are included in this published article. The code to reproduce the main results is available at https://github.com/

## Abstract

Revealing the relationship between neural network structure and function is one central theme of neuroscience. In the context of working memory (WM), anatomical data suggested that the topological structure of microcircuits within WM gradient network may differ, and the impact of such structural heterogeneity on WM activity remains unknown. Here, we proposed a spiking neural network model that can replicate the fundamental characteristics of WM: delay-period neural activity involves association cortex but not sensory cortex. First, experimentally observed receptor expression gradient along the WM gradient network is reproduced by our network model. Second, by analyzing the correlation between different local structures and duration of WM activity, we demonstrated that small-worldness, excitation-inhibition balance, and cycle structures play crucial roles in sustaining WM-related activity. To elucidate the relationship between the structure and functionality of neural networks, structural circuit gradients in brain should also be subject to further measurement. Finally, combining anatomical data, we simulated the duration of WM activity across different brain regions, its maintenance relies on the interaction between local and distributed networks. Overall, network structural gradient and interaction between local and distributed networks are of great significance for WM.

## Author summary

The Brain Connectome Project has made significant strides in uncovering the structural connections within the brain on various levels. This has led to the question of how brain structure and function are related. To further understand the relevance of structure and function in brain neural networks, we explored how WM activity duration is affected by network structure in a WM task function. Firstly, we constructed a spiking neural network and found a dependence of WM activity duration on synaptic currents. This dependence is consistent with the recent experimental observation of a gradient in receptor expression along the WM gradient network. Second, we performed the WM task

YuDong101/Structure_affects_delayed-period_activities.

**Funding:** This work is supported by National Natural Science Foundation of China (12175080 to YJ), self-determined research funds of CCNU from the colleges' basic research and operation of MOE (CCNU22JC009 to YJ), and the Central China Normal University's excellent postgraduate education innovation funding project (2023CXZZ129 to DY). The funders had no role in study design, data collection and analysis, decision to publish, or preparation of the manuscript.

**Competing interests:** The authors have declared that no competing interests exist.

independently by generating different randomized networks. It was found that network structure can be a key factor in separating persistent and non-persistent activities during the delay period. Over-expression of structures representing information transmission and cycle contributes to the maintenance of WM activity. Finally, in conjunction with anatomical data, we modeled the duration of WM activity in different brain regions. We suggest that WM-related activity relies on interactions between local and distributed networks.

## Introduction

Working memory (WM) is a crucial cognitive function responsible for temporarily storing and manipulating information, spanning fleeting moments of mere seconds. Studies have shown that association areas neurons (e.g., prefrontal cortex) exhibit selective firing activity engaged in WM tasks, which persists during the memory delay period [1–3]. In contrast, neurons in sensory cortices (e.g., V1, V2, and V4) do not seem to carry specific information during the delay period [4].

Recent neural computational models have highlighted the importance of local recurrent connections and N-methyl-D-aspartate (NMDA) receptors in encoding memory items in specific neural populations [5–8]. Experimental evidence supports these predictions [9]. WM tasks appear to rely on slow temporal dynamics, such as those involving NMDA, to maintain functionality [10]. Biophysical time constants of depolarization-induced suppression of inhibition [11], calcium-dependent nonspecific cationic current [12–15], and short-term facilitation [10,16–18] are much slower compared to NMDA receptor-mediated synaptic excitation. Although these mechanisms enhance the accuracy of memory tracking, they hinder rapid memory erasure and network reset [19]. Therefore, WM function relies on the interaction of multiple mechanisms.

Apart from examining local circuitry, recent research on large-scale networks has revealed exhilarating outcomes [20,21]. Mejías and Wang constructed and examined an anatomically constrained macaque cortical network computational model [21]. They demonstrated that internal states of WM may arise from the reverberation between regions (distributed), rather than from isolated regions alone (local). Distributed WM is more resilient to interference than local WM. Furthermore, anatomical data also suggest differences in local microcircuits [22,23]. Microcircuitry differences exist between the early sensory cortex and association cortex, stemming from the fact that the number of layer 3 neurons in the PFC has 16-fold more spines than in area V1 [22,23]. Additionally, the number of dendritic spines per unit length in PFC is fourfold greater than in V1. It would clearly result in differences in the network structure of neural circuits [4].

Despite extensive work discussing the roles of local and global mechanisms in WM, to our knowledge, regulatory mechanism of network structure on WM has yet to be validated. Indeed, network structure determines functionality [3,24–26]. Graph-theoretical studies showed that brain neural networks exhibit small-world properties [27–29]. Furthermore, small-world properties [27,30], excitatory-inhibitory ratio [31], centrality of neurons [32], and nonrandom network motifs [27,30,32] were found to be overexpressed in the cortex. Specificity of structural expression appears to be correlated with neuronal morphology [33]. As the Brain Initiative Cell Census Network project continues to develop, brain connectomes are being uncovered from multiple temporal and spatial scales [34–37]. Therefore, it is crucial to explore the relationship between brain structure and function.

In this article, we constructed a spiking neural network model in macaque cortex, examined how WM-related activity is regulated by synaptic conductance gradient and the local network structure of neural circuits. WM-related activity involves associated areas, but not sensory areas, our finding aligns with recent analyses of delay-period activity in macaque cortex [4]. The duration of delay-period activity depends on the synaptic conductance gradient, aligning with neurotransmitter gradients within WM network [38,39]. The latter was observed experimentally in the transcript [38] and receptor densities [39] of the glutamate and GABA in macaque cortex. Interestingly, a bistable region exists in synaptic parameter space, resulting from higher-order interactions within the local network structure. We explored various network structural perspectives and verified that information propagation and recurrent structures are crucial factors in maintaining WM activity. In particular, the concept offers a novel view on WM dominated by local structural gradients which was previously overlooked. Additionally, incorporating anatomical data [22,23], we simulated the duration of WM activity in different brain regions and compared with recent analyses of delay-period activity in macaque cortex [4]. It is found that maintenance of WM activity relies on interaction between local and distributed networks.

## Materials and methods

### Computational model: spiking neuronal model

Our aim was to investigate the relation between the emergence and vanishing of delayed-period firing activity and the network structure in a WM task, where the Hodgkin-Huxley (HH) neuron model was employed to model the neurons. As a class-2 neuron model, it has a discontinuous *f-I* curve (shown in S1 Fig), whose baseline activity requires a higher background noise threshold [40,41]. In contrast to class-1 neurons that fire slowly in response to weak stimuli (e.g., leaky integrate-and-fire neuron model, which has a continuous *f-I* curve), class-2 neurons easily eliminate baseline activity caused by background noise. We can focus more on the relationship between network structure and function. The membrane potential of the *i*th neuron is described as [41]:

$$C_m \frac{dV}{dt} = -g_K n^4 (V - E_K) - g_{Na} m^3 h (V - E_{Na}) - g_L (V - E_L) + I_{syn}. \tag{1}$$

The membrane capacitance $C_m$ is 0.5 nF and 0.25 nF for excitatory and inhibitory neurons, respectively. Nernst potentials for the potassium and sodium ions are $E_K$ = -80 mV and $E_{Na}$ = 40 mV, respectively. $E_L$ = -65 mV is the potential at time when leakage current is zero. $g_K$ = 4.74 $\mu$S, $g_{Na}$ = 12.5 $\mu$S, and $g_L$ = 0.025 $\mu$S are maximum conductance of potassium, sodium, and leakage currents, separately. The gating variables $n$, $m$, and $h$, which characterize average proportion of working channels opening obey the following equation:

$$\frac{dy}{dt} = \alpha_y (1 - y) - \beta_y y. \, (y = n, m, h) \tag{2}$$

The $\alpha_y$ and $\beta_y$ in Eq (2) are switch rates of ionic channels which depend on voltage and are described as follows:

$$
\begin{cases}
\alpha_n = \dfrac{0.01(V + 20)}{1 - exp[-(V + 20)/10]}, \beta_n = 0.125 \, exp[-(V + 30)/80], \\[2mm]
\alpha_m = \dfrac{0.1(V + 16)}{1 - exp[-(V + 16)/10]}, \beta_m = 4 \, exp[-(V + 41)/18], \\[2mm]
\alpha_h = 0.07 \, exp[-(V + 30)/20], \beta_h = \dfrac{1}{1 + exp[-V/10]}.
\end{cases}
\tag{3}
$$

The parameters used in the neuron model were previously reported and applied in some studies that modeled cortical neuronal populations [42,43].

## Computational model: cortical connectivity rules

Cortical network consists of a subset of $N_E = 80$ pyramidal cells and a subset of $N_I = 20$ interneurons. Each neuron receives excitatory synaptic currents from pyramidal cells and inhibitory synaptic currents from interneurons. To simplify matters and follow the previous paradigm [20], we assumed that local microcircuit is qualitatively canonical, meaning it is the same across areas, but quantitative differences in connectivity are critical. In particular, there is a dramatic increase in the number of basal dendritic spines in layer-3 of pyramidal cells from primary sensory to prefrontal areas [4]. Therefore, pyramidal cells in more highly hierarchical areas may receive more synaptic currents. In our model, pyramidal cells received contacts randomly from other neurons with varying probabilities for sensory cortex and associated cortex. In contrast, interneurons had a fixed probability of 20%. To simplify the model, we set the reception probability of pyramidal cells in sensory area to 5%, and in associated area to 20% in Fig 1A. Therefore, we can utilize such connection rules to construct the recurrent adjacency matrix $M_A^{rec}$ between nodes. The macaque brain image in the inset was drawn using anatomical data from Markov et al. quantitative measurements of macaque cortical areas [44].

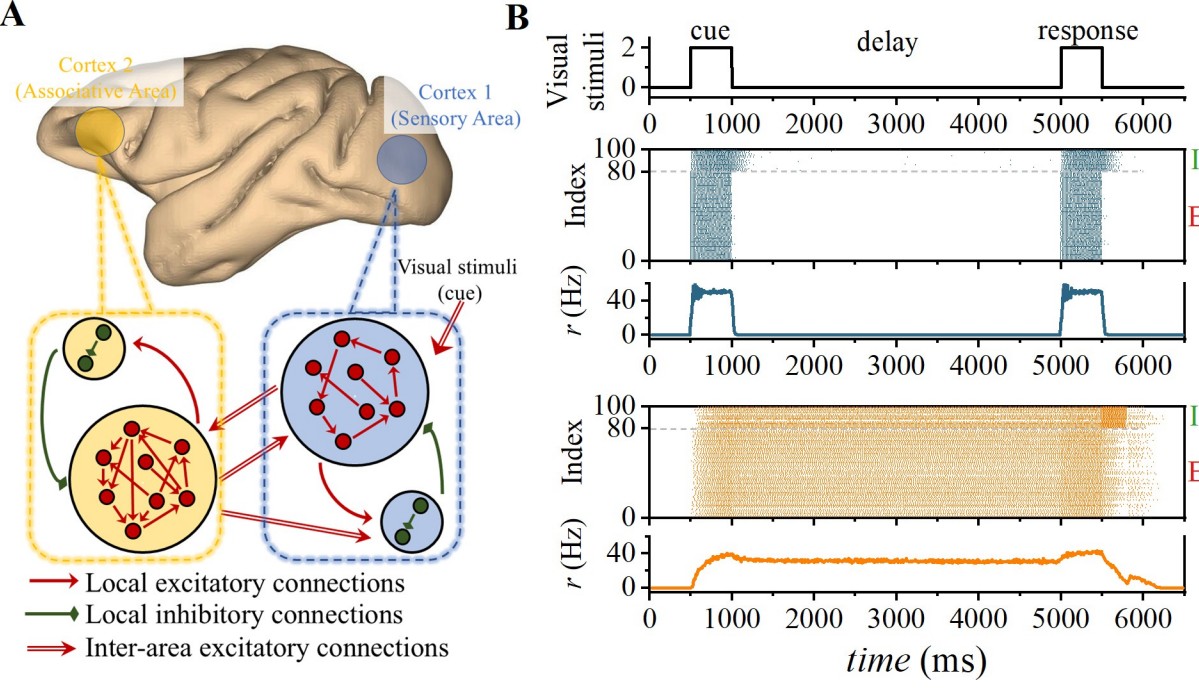

**Fig 1. Emergence of WM in spiking neural cortical networks.** (A) The model includes two regions: association cortex and sensory cortex, both modeled by the Hodgkin-Huxley neuron model (see method). (B) Spike rasters and firing rates $r$ in a delayed match-to-sample simulation. Each point represents a spike event. In the matching-to-sample simulation, visual stimuli (upper panel) are modeled as suprathreshold signals applied to excitatory populations in the sensory cortex. During the delay period, WM-related activity involves the associative cortex rather than primary sensory cortex. Synaptic conductances were set to $g_{NMDA}$ = 0.13 μS, $g_{AMPA}$ = 0.11 μS, and $g_{GABA}$ = 0.47 μS. The amplitude of the excitatory presynaptic currents of NMDA and AMPA were 19 pA and 71 pA, respectively, within the range of electrophysiological data [54,55]. The NMDA/AMPA peak ratio is 26.6% consistent with what was observed experimentally [54].

Regarding the interregional connections, we assume a well-defined hierarchical relationship between two areas (e.g., areas V1 and PFC). Along the visual hierarchy, forward-feed projections originate from supragranular layers, primarily targeting layer 4 and subsequently projecting to layer 2/3 of target area [45–47]. In our model, it is approximated as a projection from pyramidal cells in sensory cortex to pyramidal cells in associative cortex. Feedback projections are more diffuse on their targets compared to forward-feed projections [46]. We hypothesize that feedback projections originate from pyramidal neurons in associative cortex and target all populations in sensory cortex. Therefore, as shown in Fig 1A, the projection from the association area to the sensory area (feedback) targets pyramidal cells and interneurons, whereas the projection from the sensory cortex to the association cortex (feedforward) targets only the pyramidal cells. Constrained by the number of basal dendritic spines in layer-3 of pyramidal cells, cross-area connections were similar to intra-area connections, with the reception probability set at 5% for pyramidal cells in the sensory area and 20% in the associated area. Reception probability of interneurons in the sensory area is fixed at 5%. Thus, we employ such connectivity rules to construct the adjacency matrix $M_A{}^{ext}$ for interregional interactions.

## Computational model: synaptic dynamics

Within neuronal connections, there are two distinct types of synapses: excitatory synapses (glutamatergic) and inhibitory synapses (GABAergic), which are respectively projected by pyramidal cells and interneurons. Local recurrent excitatory postsynaptic currents (EPSCs) consist of two components, which are mediated by AMPA and NMDA receptors. Conversely, the cross-area external EPSCs are exclusively governed by AMPA receptors [7]. Total synaptic currents are given by:

$$I_{syn}^i = I_{\text{AMPA},ext}^i + I_{\text{AMPA},rec}^i + I_{\text{NMDA},rec}^i + I_{\text{GABA},rec}^i \tag{4}$$

in which

$$I_{\text{AMPA},ext}^i = g_{\text{AMPA},ext}(V_i - V_E) \sum_j^{N_E} \varepsilon_{i,j}^{ext} s_j^{\text{AMPA},ext}(t), \tag{5}$$

$$I_{\text{AMPA},rec}^i = g_{\text{AMPA}}(V_i - V_E) \sum_j^{N_E} \varepsilon_{i,j}^{rec} s_j^{\text{AMPA},rec}(t), \tag{6}$$

$$I_{\text{NMDA},rec}^i = \frac{g_{\text{NMDA}}(V_i - V_E)}{1 + [\text{Mg}^{2+}] exp(-0.062V_i)/3.57} \sum_j^{N_E} \varepsilon_{i,j}^{rec} s_j^{\text{NMDA},rec}(t), \tag{7}$$

$$I_{\text{GABA},rec}^i = g_{\text{GABA}}(V_i - V_I) \sum_j^{N_I} \varepsilon_{i,j}^{rec} s_j^{\text{GABA},rec}, \tag{8}$$

where $V_E$ = 0 mV, $V_I$ = -70 mV. The term $\varepsilon_{i,j}{}^{rec}$ and $\varepsilon_{i,j}{}^{ext}$ is a matrix element in adjacency matrix $M_A{}^{rec}$ and $M_A{}^{ext}$ of cortical network. $N_E$ and $N_I$ denote subsets of excitatory and inhibitory neurons, respectively. $\varepsilon_{i,j}$ = 1 if neuron $i$ is coupled to neuron $j$ and $\varepsilon_{i,j}$ = 0 otherwise. Recurrent NMDA currents are voltage-dependent and their activity is regulated by extracellular magnesium concentration $[\text{Mg}^{2+}]$ = 1 mM [48]. The gating variables $s$ that determine the proportion of open channels are described as follows. The AMPA (external and recurrent) channels are regulated by

$$\frac{ds_j^{\text{AMPA}}(t)}{dt} = -\frac{s_j^{\text{AMPA}}(t)}{\tau_{\text{AMPA}}} + \sum_k \delta(t - t_j^k) + D_p \xi(t). \tag{9}$$

And the NMDA channels are controlled by

$$\frac{ds_j^{\text{NMDA}}(t)}{dt} = -\frac{s_j^{\text{NMDA}}(t)}{\tau_{\text{NMDA},decay}} + \alpha x_j(t) \sum_k (1 - s_j^{\text{NMDA}}(t)),$$  (10)

$$\frac{dx_j(t)}{dt} = -\frac{x_j(t)}{\tau_{\text{NMDA},rise}} + \sum_k \delta(t - t_j^k).$$  (11)

Lastly, the GABA channels are described by

$$\frac{ds_j^{\text{GABA}}(t)}{dt} = -\frac{s_j^{\text{GABA}}(t)}{\tau_{\text{GABA}}} + \sum_k \delta(t - t_j^k),$$  (12)

the term $\tau_{\text{AMPA}} = 2$ ms, $\tau_{\text{NMDA},rise} = 2$ ms and $\tau_{\text{NMDA},decay} = 100$ ms are time constant of AMPA and NMDA currents [49,50]. The time constant of GABA currents is $\tau_{\text{GABA}} = 10$ ms [51,52]. The synaptic conductance across the region was fixed to $g_{\text{AMPA},ext} = 0.22$ μS. The sum over $k$ represents the sum over spikes emitted by presynaptic neuron $j$. In addition to external AMPA currents originating from neurons in source area, neurons are also subjected to spiking inputs from other background neurons, which was modeled in Eq (9) as synaptic noise $\xi(t)$ generated by a Poisson process with rate $\nu_{\text{ext}} = 2300$ Hz and intensity $D_p = 0.005$, independently from neuron to neuron.

## Multiple regression analysis

Multiple regression analysis is a statistical method used to explain the relationship between a dependent variable (duration of WM-related activity) and multiple independent variables (topological structures). Through multiple regression, we can identify independent variables that have statistically significant effects on the dependent variable and quantify the extent of their impact. Since duration is not linearly dependent on network structure, we need to apply a nonlinear transformation to dependent variable (*Duration*):

$$ln(Duration) = \beta_0 + \sum_i \beta_i X_i.$$  (13)

The vector $\beta$ represents regression coefficients, which quantify the influence of each structural variable $X_i$ on the duration of WM activity. To address multicollinearity (cross-correlation) among the structural variables (as shown S3 Fig), we employ ridge regression. In ridge regression, cost function is formulated as the weighted combination of ordinary least squares loss function and the L2 regularization term. For a dataset with $n$ samples and $p$ independent variables, the cost function can be expressed as follows:

$$\text{Cost}(\beta) = \frac{1}{n} \sum_{i=1}^{p} (y_i - \beta^T X_i)^2 + \lambda \|\beta\|_2^2.$$  (14)

The vector $\beta$, obtained by minimizing cost function, represents the regression coefficients in ridge regression. The regularization parameter $\lambda$ determined through cross-validation. In ridge regression, when there is multicollinearity among the independent variables, the signs of $\beta$ may become unreliable. However, magnitude of $\beta$ can still be used to assess importance of independent variables. Therefore, we utilize the absolute values of $\beta$ to evaluate the significance of independent variables.

## Small world organization

Path length between two nodes in a network is represented by number of connections along that path. The shortest path $l_{i,j}$ refers to the path with minimum number of connections. To extend the characteristic in entire network, we calculated all SPL in the network, and obtained average SPL $\langle l \rangle$ between all pairs of neurons.

$$\langle l \rangle = \frac{1}{n(n-1)} \sum_{i,j;i \neq j} l_{i,j}. \tag{15}$$

Clearly, the metric is well-defined in any connected network where it is possible to reach any node from any other node.

Clustering coefficient ($Cc$) characterizes the tendency of nodes to cluster together. Local clustering measure of a single node quantifies the level of interconnectedness among its neighbors. We can obtain local clustering coefficient of node $i$ by using its adjacency matrix $M_A^{rec}$:

$$Cc_i = \frac{2E_i}{k_i - (k_i - 1)}, \tag{16}$$

where $E_i$ represents the number of edges among neighboring nodes of node $i$, and $k_i$ represents degree of node $i$. The clustering coefficient $<Cc>$ for network is calculated by taking the average of all local clustering coefficients.

$$\langle Cc \rangle = \frac{1}{n} \sum_i Cc_i. \tag{17}$$

## Eexcitatory-inhibitory ratio

In network science, number of postsynaptic current received by a neuron is termed as in-degree $d$. The terms $d_{\text{E to E}} = \sum_i^{N_E} \sum_j^{N_E} \varepsilon_{i,j}$ and $d_{\text{I to I}} = \sum_i^{N_I} \sum_j^{N_I} \varepsilon_{i,j}$ represent synaptic connections within excitatory and inhibitory populations, while $d_{\text{E to I}} = \sum_i^{N_E} \sum_j^{N_I} \varepsilon_{i,j}$ and $d_{\text{I to E}} = \sum_i^{N_I} \sum_j^{N_E} \varepsilon_{i,j}$ denote interactions between the two populations. The ratio between them can be used to quantify degree of interaction among E/I neurons:

$$F_{E/I \text{ balance}} = \frac{d_{\text{E to E}} + d_{\text{I to I}}}{d_{\text{E to I}} + d_{\text{E to I}}}. \tag{18}$$

## Eexcitatory and inhibitory hub

The proportion of excitatory input at $i$-th node is:

$$P_i = \frac{k_i^E}{k_i^E + k_i^I}. \tag{19}$$

$k_i^E$ and $k_i^I$ denote in-degree received by $i$-th pyramidal cells from excitatory (originating from pyramidal cells) and inhibitory (originating from interneurons) inputs, respectively. Neurons with a $P_i$ value greater than 80% were identified as hubs. The proportion of excitatory hubs

and inhibited hubs at network level was:

$$P_{E,hub} = \frac{N(P_i > 80\%)}{N_{nod}} \text{ for } i \in N_E. \tag{20}$$

$$P_{I,hub} = \frac{N(P_i > 80\%)}{N_{nod}} \text{ for } i \in N_I. \tag{21}$$

Considering the opposing roles of excitatory hubs and inhibitory hubs in network, we use their ratio to measure the relative expression level of excitatory hubs in network:

$$F_{\text{rich hub}} = \frac{P_{E,hub}}{P_{I,hub}}. \tag{22}$$

### Local recurrent cycles

Cycle can be simply defined as a closed path with same starting and ending nodes. It is simplest structure that introduces redundant paths in network connections and feedback effects in network dynamics. Size of cycles is equal to number of links (or nodes) it contains. The number of cycles increases exponentially with size (see Results). Therefore, considering the enormous computational complexity, it is not practical to cover all cycles. However, number of cycles is strongly correlated between different sizes (see Results). In other words, if there are many cycles with a girth of 3, then there are also many cycles with girths of 4, 5, and 6 in network. We can measure the frequency of cycles in a network by counting the number of cycles with a girth of 3.

### Nonrandom motifs

A motif is a connected graph or network composed of $M$ vertices and a set of edges, used as a subgraph to form a larger network. For a directed graph, the maximum number of edges is $M^2$—$M$, while the minimum is $M$—1, ensuring connectivity. For $M$ = 3, number of corresponding motif classes is 13 [53]. A larger network can be structurally assembled from a finite set of such motifs. Essentially, structural motifs serve as foundational building blocks for larger networks. We recorded occurrence of each motif (see S5 Fig) in network and observed the correlation between sustained WM-related activity and motif frequencies.

## Results

In our cortical spiking neural network model, the WM tasks [7] were simulated using the following delayed response protocol: (1) Start the simulation with a pre-cue interval of 500 ms, during which the sensory and associative areas exhibit silent activity. (2) The stimulus presentation (cue) involves supra-threshold (threshold is 0.7 nA, see S1 Fig) stimulation of the pyramidal cells in the sensory area, lasting for 500 ms, as depicted in Fig 1B. (3) Following the removal of the external stimulus, a delay period of 4000 ms ensues. (4) Finally, a matching stimulus appears within 500 ms, using the same intensity. Within 300 ms after the presentation of the matching stimulus, the interneurons of associative areas receive a stimulus of the same intensity to account for the increased input due to behavioral response/reward signals.

In Fig 1B, neural activity persists in the associated area rather than the sensory area, aligning with the findings observed in the experiment [4]. In our model, the primary distinction between sensory cortex and association cortex lies in the probabilities of excitatory

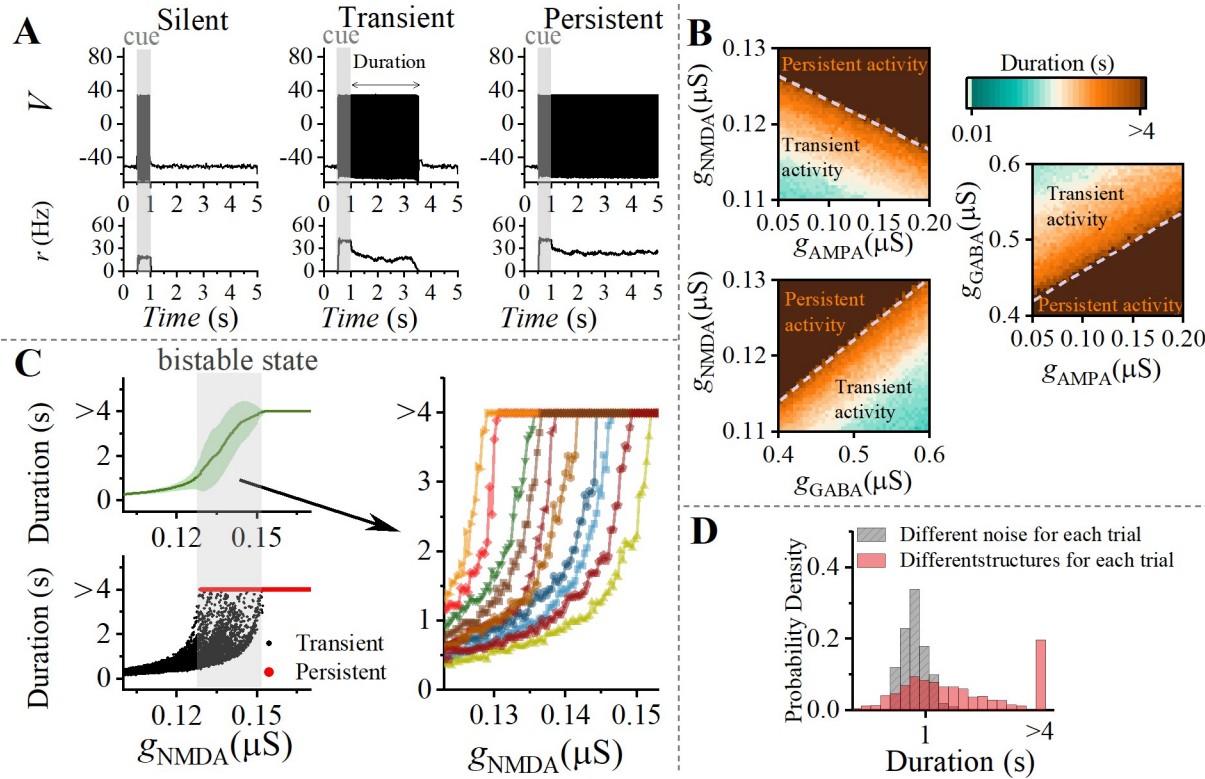

**Fig 2. Synaptic mechanism of WM-related activity.** (A) The panel displays representative WM activity under three different synaptic parameters. Network responses to the cue signal (gray bar at 0.5-1s) are represented as follows: silent (left, $g_{NMDA}$ = 0.05 μS), transient (middle, $g_{NMDA}$ = 0.13 μS), and persistent (right, $g_{NMDA}$ = 0.15 μS). The time from cue removal until the last neuron returns to silence is defined as the duration of WM activity. (B) Duration of WM activity is influenced by synaptic conductance $g_{NMDA}$, $g_{AMPA}$, and $g_{GABA}$. Specifically, the duration of WM is positively correlated with $g_{NMDA}$ and $g_{AMPA}$, and negatively correlated with $g_{GABA}$, aligning with experimental findings [38,39] (details in the text). (C) During the transition from transient to sustained states, bistable states emerge as a result of higher-order interactions within the network structure. Top left panel: The dependency of duration on $g_{NMDA}$ across 50 different network structures. The green solid line represents the mean value, while the green shaded area represents the standard deviation. Bottom left panel: A bistable region exists in the transition from transient to sustained states. Right panel: Examples of eight representative networks. The network structure influences the transition threshold of $g_{NMDA}$. (D) Distribution of duration of WM activity in the context of random network structure and random noise, for a fixed set of parameters. Randomness of noise leads to a Gaussian distribution of duration (gray), while heterogeneity of the topology structure results in a bimodal distribution of duration (red).

postsynaptic currents received by pyramidal cells (see in the method). This hypothesis is based on varying numbers of basal dendritic spines observed anatomically in layer 3 pyramidal cells across different brain regions (e.g., V1 has a count of 643, while 9/46d has a count of 7800) [22,23]. Consequently, our results demonstrate a correlation between the gradient of synaptic spines and maintenance of delayed-period activity in WM task.

We are interested in what factors influence the duration of WM activity in local areas. During the delay period, neural activity can be categorized into three distinct types. (i) After presentation of WM cue, pyramidal cells quickly return to a resting state (where the information of WM cannot be retained), as shown in left panel of Fig 2A. (ii) A limited number of population spikes occur before returning to resting state, as shown in middle panel of Fig 2A. (iii) Sustained spiking state persists until the matching stimulus appears, as shown in right panel of Fig 2A. In subsequent discussions, we focus on how synaptic conductance gradients and network structure impact the duration of WM activity.

## Synaptic conductance gradient regulates WM-related activity

Recent experiments conducted in the neocortex of humans and macaque revealed that the glutamate and GABA neurotransmitters in layer 3 exhibit opposite V1-V2-PPC-PFC gradients across regions of the WM network [38]. It was found that Glutamate markers are lower in V1 (sensory cortex) and higher in PFC (association cortex), while GABA markers are higher in V1 and lower in PFC.

As illustrated in Fig 2B, we demonstrated the dependence of WM duration on the intensity of synaptic current receptors (NMDA, AMPA and GABA). WM duration is directly proportional to $g_{NMDA}$ and $g_{AMPA}$, while inversely proportional to $g_{GABA}$. The gradient in neurotransmitter receptor expression facilitates rapid and reliable information processing in sensory cortical areas, while supporting slow and flexible integration in higher cognitive areas [39]. Based on experimental observations and our findings, it can be further established that the distribution of WM across the brain correlates with the gradient of neurotransmitters.

In Fig 2C, we explored the relationship between the duration and $g_{NMDA}$ across 50 different network structures. Within a specific parameter range, a significant standard deviation is observed during the transition from transient to persistent states (upper left panel). When examining the distribution of duration across these 50 networks (bottom left panel), it is evident that a bistable state exists during the transition between transient and persistent states. Further investigation reveals that network structure influences the transition threshold (right panel). The bistable state induced by network structure was also observed in phase oscillators [56,57] and epidemiological models [58]. In summary, the maintenance of delayed-period activities is not only related to synaptic mechanisms, but also to the topological structure of neural networks.

## The primary topological structures that drive WM-related activity

There are two sources of randomness in our model: one is the spiking inputs that neurons receive from other background neurons, as described in Eq (9), and the other is the randomness in the network structure. In the following, we fixed the synaptic conductance to $g_{NMDA}$ = 0.13 µS, $g_{AMPA}$ = 0.11 µS and $g_{GABA}$ = 0.47 µS. In trials where the network structure is fixed but the spiking inputs are random (grey distribution in Fig 2D), the duration follows an unimodal Gaussian distribution. Conversely, in trials where the spiking inputs are fixed but the network structure is randomized (red distribution in Fig 2D), the duration follows a bimodal distribution. It should be emphasized that the source of the bistability of system is the network structure.

## Excitation-inhibition balance

The balance between excitatory and inhibitory synaptic inputs (referred to as E/I balance) represents a fundamental principle governing neural dynamics and computational functionality in cortical circuits [59]. Disruption in E/I balance in the cortex can result in various cognitive deficits [60]. For the associated cortex, synaptic connections between neurons are regulated by recurrent adjacency matrix $M_A^{rec}$ (see Method). To examine the impact of E/I balance in associated cortex on the maintenance of delayed-period activities, we divided adjacency matrix into four regions, as shown in left panel of Fig 3A. Results in right panel indicate that duration is positively correlated with $d_{E\ to\ E}$ and $d_{I\ to\ I}$, while negatively correlated with $d_{E\ to\ I}$ and $d_{I\ to\ E}$. Further details can be verified by Fig 3B.

A larger $d_{E\ to\ E}$ indicates more strong recurrent excitation mediated by NMDA receptors, promoting population excitation [2,7,59]. However, firing activity of GABAergic inhibitory interneurons leads to a decrease in overall excitability of the area. Therefore, a large $d_{I\ to\ I}$ leads

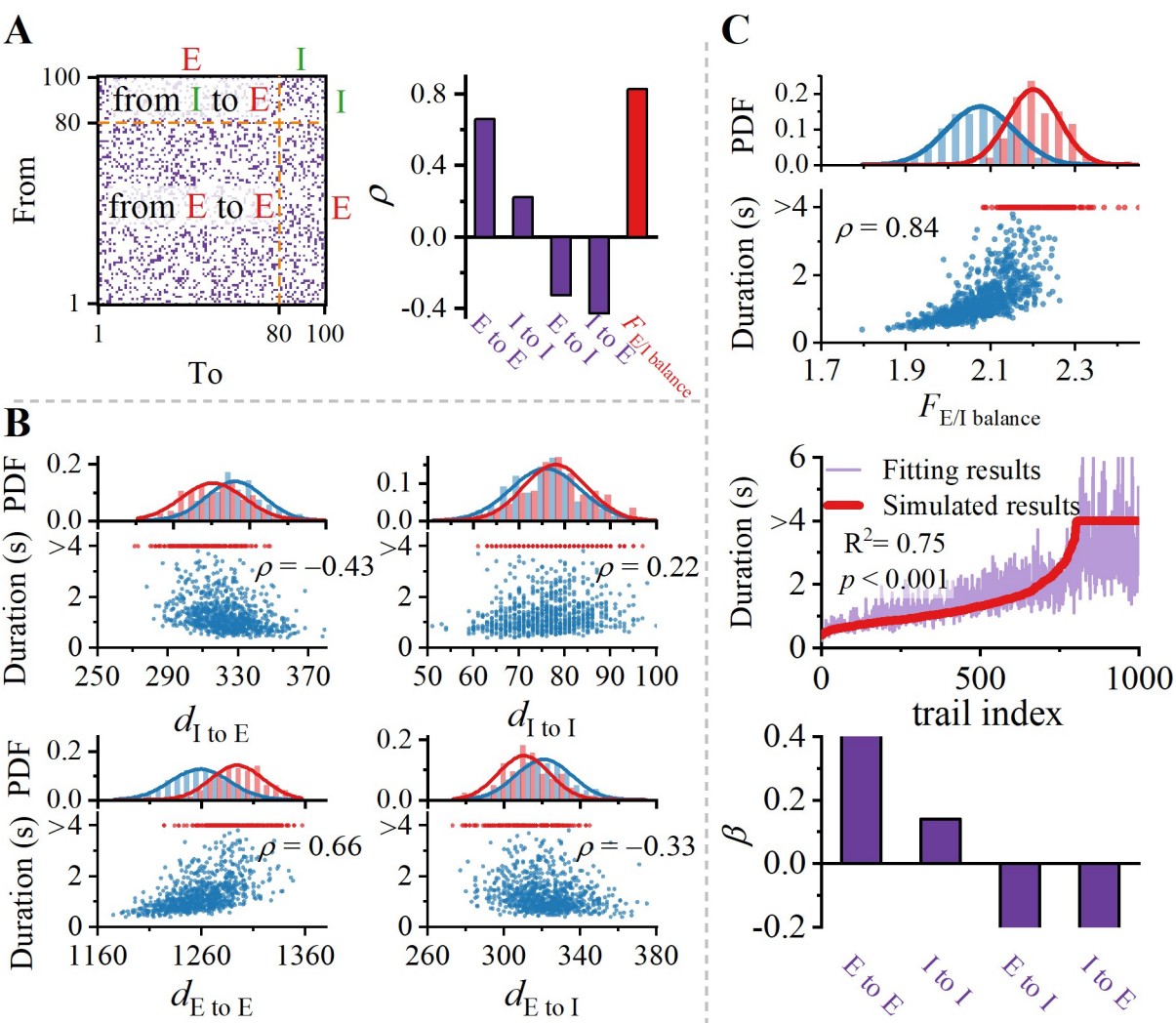

**Fig 3. Excitation-inhibition balance of network regulates the duration of WM.** (A) Left panel: Scheme of four intra-area synaptic connectivity patterns. Right panel: Spearman correlation coefficient $\rho$ between degree of intra-area connectivity $d$ (detail see method) and duration of WM activity. (B) Dependence of duration of WM activity on $d$. Spearman correlation coefficients $\rho$ are depicted ($p < 0.001$ for all). (C) Upper panel: Duration is strongly correlated with E/I balance. Middle panel: Multivariate regression of degree of four synaptic connections types on duration of WM activity. Red line represents our arranged simulated data, and purple line represents corresponding fitted data. There is no collinearity among structural variables (see S3 Fig). Lower panel: The regression coefficients for the four synaptic types.

to self-inhibition of inhibitory populations, which in turn promotes the excitation level of excitatory populations. A natural explanation for this phenomenon is disinhibition [61, 62]. As a result, the correlation is positive. The terms $d_{\text{E to I}}$ and $d_{\text{I to E}}$ activate inhibitory populations, which in turn inhibit excitatory populations, creating a pathway for lateral inhibition, which results in negative correlation. To characterize the levels of E/I balance within the region, we introduced a novel factor, the E/I balance factor $F_{\text{E/I balance}}$ (see Methods). It is significantly enhances the correlation, as depicted in upper panel of Fig 3C.

Multiple regression analysis also validates our results. In the middle panel of Fig 3C, reorganized simulation data in ascending order of duration are compared with fitted curve obtained through multiple regression. Fitted curve closely matches simulated results when duration is less than 4000 ms. However, when duration approaches or exceeds 4000 ms, results of the multiple regression struggle to accurately predict simulated results. Two primary reasons account

for the discrepancy: firstly, simulated results do not conform to the characteristics of a specific curve, making it challenging to find a single equation to fit them; secondly, E/I balance alone is insufficient to sustain WM, as other latent variables come into play. Existence of a bistable region in the delayed-period activities at single-cell scale further confirms the latter (see S2 Fig).

Nonetheless, the results of regression coefficients in lower panel of Fig 3C are qualitatively consistent with correlation coefficients. The magnitude and sign of regression coefficients represent the extent and direction of independent variable's impact on dependent variable. Therefore, we conclude that maintenance of delayed-period activities in associative cortex is governed by the interaction of recurrent excitation mediated by NMDA receptors, disinhibition mediated by GABA, and lateral inhibition mediated by both NMDA and GABA.

## Excitation and inhibition hub

Convergence (number of presynaptic neurons) and divergence (number of postsynaptic neurons) of neurons have a profound influence on their role in facilitating information flow within a network. Neurons with high numbers of incoming or outgoing connections serve as hubs for information integration and transmission, referred to as hub neurons. As shown in Fig 4A, in WM tasks, neurons with a higher E/I ratio exhibit greater firing rates. Furthermore, neurons with higher spike generation rates (greater firing rates) have a more significant impact on postsynaptic neurons. These neurons serve as both excitatory (pyramidal cells) and inhibitory (interneurons) hubs, influencing the collective behavior of the network. These findings are robust across 1000 different network structures. Fig 4B depicts two typical examples. We categorized pyramidal neurons as excitatory hub neurons if their proportion of excitatory inputs exceeded 80%. It is evident that subject-1 has a greater number of excitatory hub neurons compared to subject-2. Therefore, subject-1 is able to sustain WM, while subject-2 is not. Hub neurons generate spikes rapidly, leading to a more pronounced impact on postsynaptic neurons.

To investigate whether delayed-period activities depend on the balance between excitatory and inhibitory hub neurons, we defined the proportion of hubs for pyramidal cells ($P_{E, Hub}$) and interneurons ($P_{I, Hub}$) within associative cortex. In Fig 4C, it is observed that WM duration is positively correlated with $P_{E, Hub}$ and negatively correlated with $P_{I, Hub}$. Therefore, maintenance of delayed-period activities also relies on the balance between excitatory and inhibitory hub neurons, which further provides evidence for the importance of E/I balance in WM. As brain connectomics continues to advance [34–37], our predictions could potentially be validated through experiments at the level of whole-cell connectivity.

## Recurrent cycle

Cycles are the simplest structures in network dynamics for introducing feedback effects and providing redundant pathways for all involved nodes [63]. The redundancy leads to intricate interactive dynamics feedback, enhancing robustness [64] and information transmission efficiency [65, 66] in the neural system. Furthermore, spine-count data implies that there are more recurrent excitatory connections in PFC (association cortex) than in V1 (sensory cortex) [4,22,23]. Mechanism of sustaining persistent activity in absence of external stimuli is believed to involve sufficiently strong recurrent excitatory connections [67,68]. Therefore, it is crucial to study the impact of cycle structures on maintaining WM.

The number of cycles grows exponentially with the girth of cycle, as shown in upper panel of Fig 5A. Although it is not feasible to calculate all cycles due to the computational

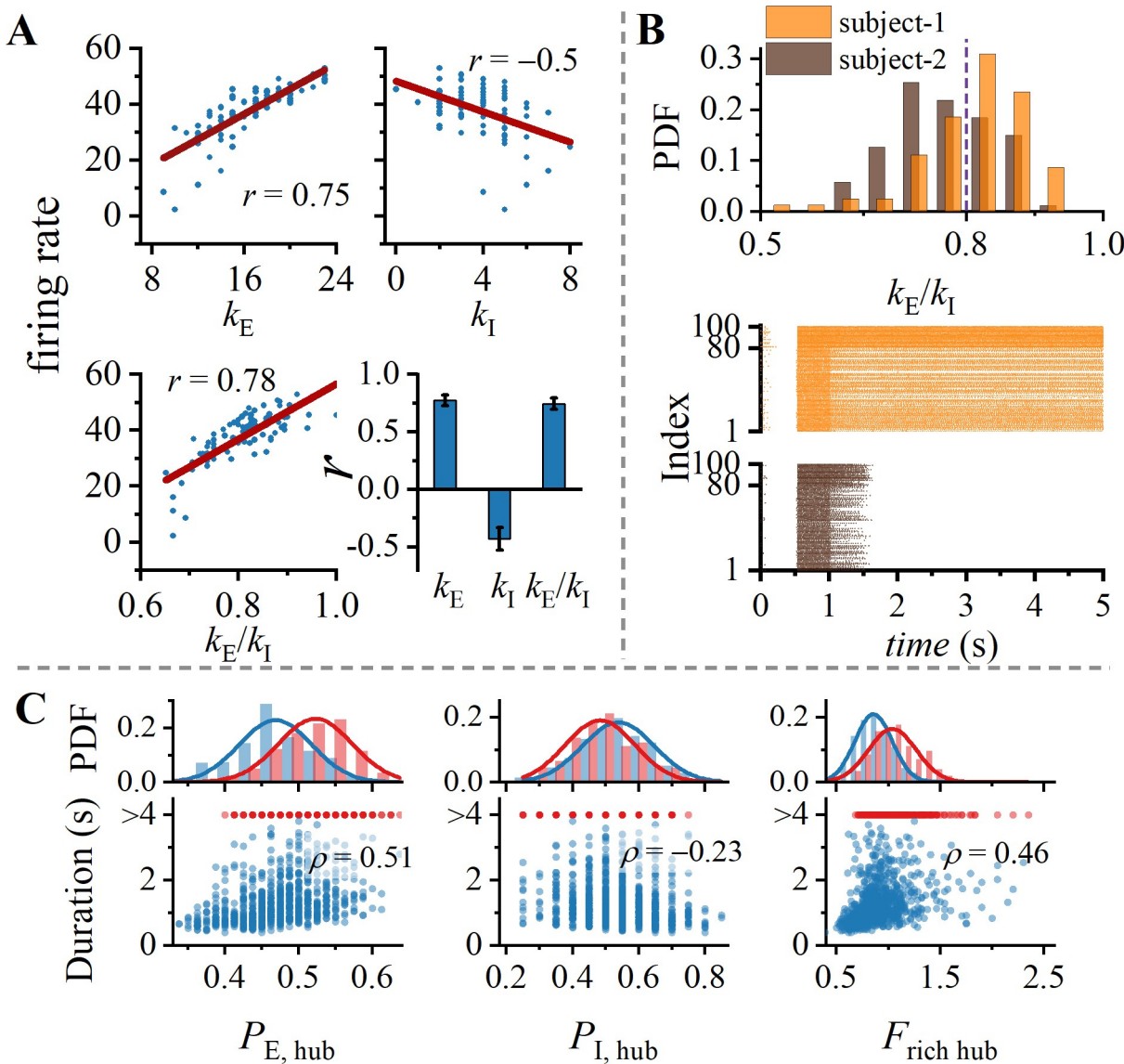

**Fig 4. Rich hub regulates duration of WM.** (A) Evidence of E/I ratio modulation on neuronal firing rates at the single-neuron scale. In a representative network, excitatory in-degree $k_E$, inhibitory in-degree $k_I$, and the ratio of both, exert regulatory effects on the duration of WM activity (all $p < 0.001$). Clearly, neurons with a higher E/I ratio exhibit greater firing rates. (B) Excitatory hub sustain WM activity during delay-period. Upper panel: Distribution of excitatory-inhibitory input ratio for pyramidal cells in two representative networks. Lower panel: Spike rasters for two representative networks. (C) Dependency of the duration of WM activity on rich hub. It is calculated from three perspectives: excitatory hub, inhibitory hub, and their ratio. Spearman correlation coefficients are depicted ($p < 0.001$ for all).

complexity, there is a strong linear correlation between counts of cycles with different girths. We measured the level of cycle structure in network by counting cycles with the girth of 3.

The calculation of cycle counts includes not only entire network but also excitatory subnetwork since excitatory and inhibitory neurons play different roles in sustaining WM. Dependence of the duration of WM activity is shown in the upper panel of Fig 5B. Cycle structure of entire network is unrelated to the duration. However, cycle structure in excitatory subnetwork is positively correlated with duration, which further confirms that excitatory recurrent plays a crucial role in maintaining WM [69]. Lack of correlation between cycle structure in entire network and duration can be attributed to inhibitory interneurons involved in cycles, disrupting

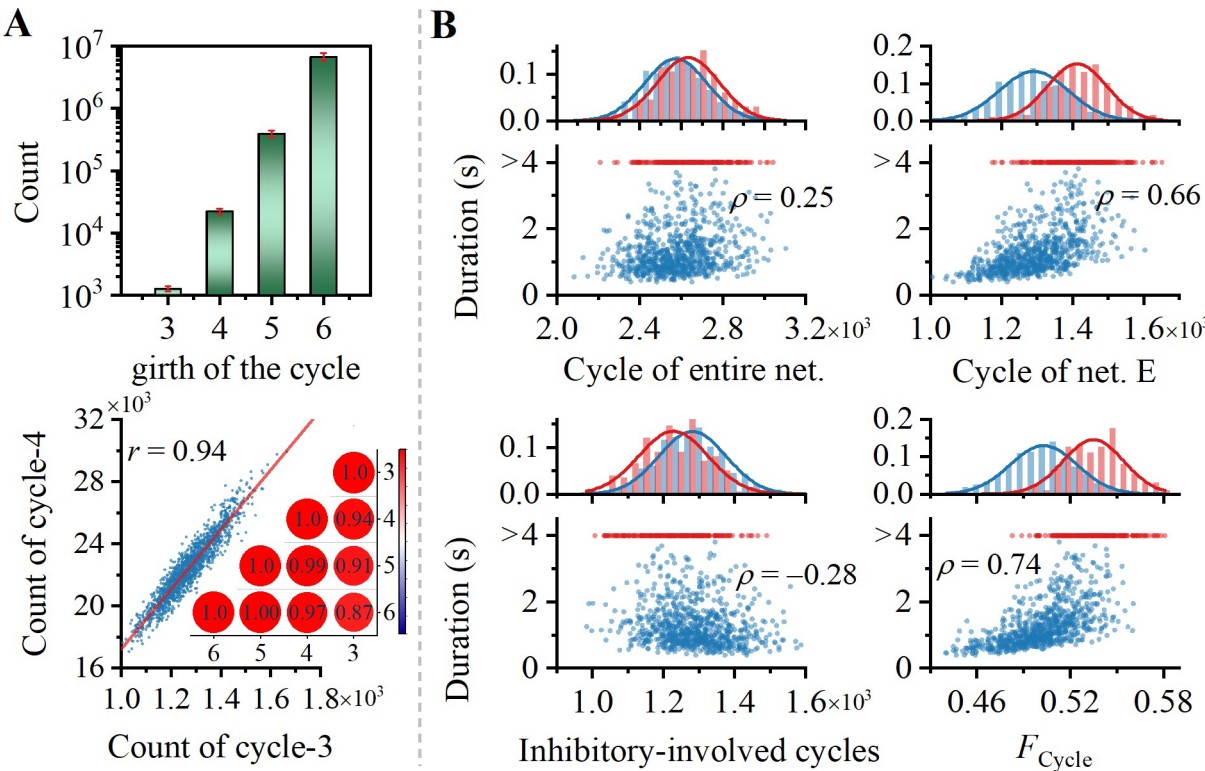

**Fig 5. Cycle structure regulates duration of WM.** (A) Upper panel: Count of cycles with different lengths in a log-linear plot. Lower panel: Cycles with different lengths exhibit a strong correlation. (B) Dependency of duration of WM activity on cycle structure. It is calculated from four perspectives: entire network, excitatory subnetwork, inhibitory involved cycles, and ratio between the latter two. Spearman correlation coefficients are depicted ($p < 0.001$ for all).

the feedback pathway. To validate this hypothesis, in lower-left panel of Fig 5B, we examined the correlation between duration and cycles structure involving the interneurons. The negative correlation between them suggests that the involvement of interneurons in cycles hinders maintenance of WM.

WM also relies on the interaction between fully excitatory cycles and inhibitory-involved cycles. Therefore, in lower right panel of Fig 5B, there is a stronger correlation between the ratio ($F_{Cycle}$) of fully excitatory cycles to inhibitory-involved cycles and duration. Furthermore, analysis of anatomical data also indicates a reduced expression of cycle structure within the visual cortex of rats [27] and macaques [70], which are both significant species within the scope of studying WM paradigms. Thus, the lack of cycle structure may underlie the absence of WM-related sustained activity in early visual cortex.

Furthermore, brain networks have a small-world organization across multiple scales [27–30]. Small-world networks exhibit characteristics of short path lengths and high clustering. [71]. In neural networks, the presence of small-worldness reflects a balance between integrating global information and maintaining local segregation [72]. The reduction of small-worldness indicates a potential decrease in information exchange efficiency and associative memory capacity [25]. Duration is positively correlated with clustering coefficient (Cc) and negatively correlated with shortest path length (SPL). Notably, small-worldness is characterized by high Cc and short SPL (shown in S4 Fig). It suggest that maintenance of delayed-period activities relies on the small-worldness of excitatory subnetwork, while the involvement of the inhibitory subnetwork would undermine it.

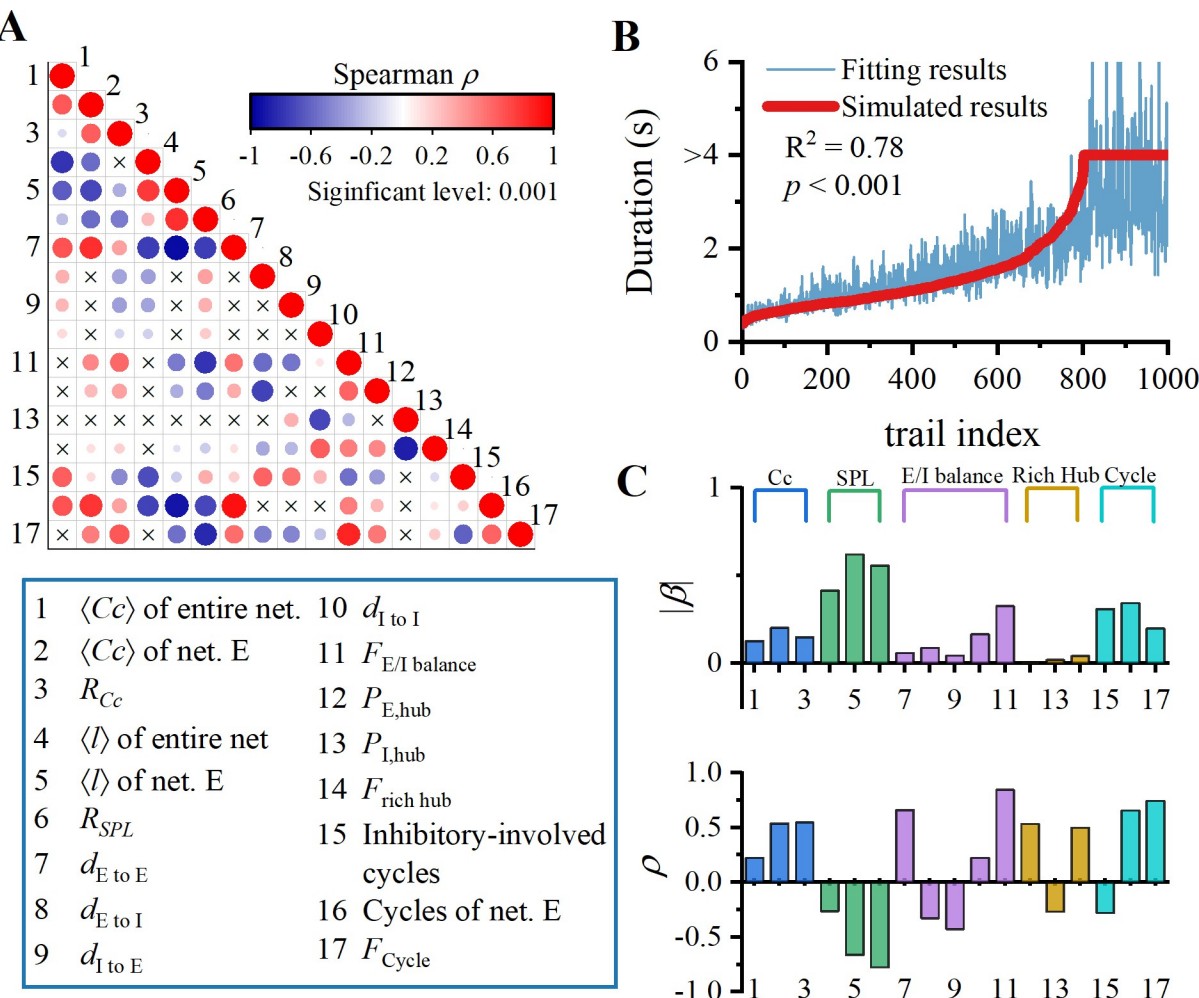

**Fig 6. How topological structures affect WM activity.** (A) Correlation matrix of seventeen structural variables. Size of dots represents correlation between variables. Red color indicates positive correlation, while blue color indicates negative correlation. (B) Multivariate regression of 17 structural variables on the duration of WM activity. Red line represents our arranged simulated data, and blue line represents corresponding fitted data. (C) Regression coefficients $\beta$ and spearman correlation coefficient $\rho$ for the 17 structural variables. The regression coefficients of small world, E/I balance, and cycle structure are larger, while the rich hubs are relatively smaller.

In Figs 3–5, we discussed effects of seventeen different network structures on maintenance of WM (excluding cycles of entire network that are unrelated to the duration). For clarity, we also assigned the number from 1 to 17 to represent these variable in lower panel of Fig 6A. These factors exhibit multicollinearity, and the correlation matrix between structural variables is depicted in Fig 6A. For instance, there is a strong positive correlation between $d_{E\ to\ E}$ and $\langle Cc \rangle$ of net. E, while there is a strong negative correlation between $d_{E\ to\ E}$ and $\langle l \rangle$ of net. E. Clearly, it is a characteristic of random networks, where increasing edge leads to increased clustering and reduced communication distance between nodes.

Fig 6B presents the results of multiple regression analysis between these 17 structural variables and WM duration. It is evident that including more structural factors in regression model can improve goodness of fit (increase in R-squared) compared to Fig 3. Furthermore, regression coefficients in Fig 6C indicate that small-worldness, E/I balance and cycle structure are the most crucial indicators for maintaining WM, while rich hub has a relatively weak

impact. It is independent of the correlation between structural variables and duration (Fig 6C, lower panel).

## Nonrandom motifs

The interaction between three neurons is referred to as triad motifs [53]. According to the definition proposed by Milo et al. [53], set of three nodes can give rise to 13 unique configuration: three motifs with two edges (motifs 1–3), four motifs with three edges (motifs 4–7), five motifs with five edges (motifs 8–12), and a fully connected motif (motif 13). Previous experimental studies discussed triad motifs in cortical network [27,30,32,33,70]. We performed a statistical analysis to determine frequency of triad motifs occurring in 1000 distinct topological structures of associated cortical network. Additionally, we examined their correlation with WM duration in Fig 7A.

Analyzing the entire network can reveal a weak correlation between the expression of all motifs and WM duration. However, when we specifically consider excitatory subnetwork, the correlation significantly increases. The rationale is analogous to the cycle structure depicted in Fig 5, where incorporation of inhibitory interneurons hinders the maintenance of delayed-period activities in motifs. Negative correlation between number of inhibitory-involved motifs and duration provides further evidence for this assertion. Indeed, the sum of expressions for motifs 7, 10, 12, and 13 represents the degree of expression for cycle structures within network (discussed in Fig 5).

However, when we calculate the ratio ($F_{\text{motifs}}$) of excitatory motifs to inhibitory-involved motifs, we do not observe significant enhancement in the overall correlation (Fig 7A, right panel), which means that regulatory effects of different motifs may vary. Through multiple regression analysis using $F_{\text{motifs}}$, we identified motifs 1, 2, 6, 7 and 10, as factors with larger regression coefficients. Therefore, motifs 1, 2, 6, 7 and 10 are more crucial for maintaining delayed-period activities. Motifs 7 and 10 introduce feedback effects in network and provide redundant paths for all relevant nodes, while motifs 1, 2 and 6 establish connectivity pathways between subregions. Our results suggest that mechanism underlying delayed-period activity in local neural circuits may involve construction of modular redundant cycles and formation of robust pathways between modules.

Previous studies have suggested that some cognitive computational functions may occur at the motifs level [73–76]. Certain motifs can serve as inputs of long-term memory and thus play an important role in many cognitive processes. In addition, localized computation dominated by the motifs is robust to changes such as the loss of a single neuron, as other neurons can be recruited into the motifs to replace any lost neurons [76]. Thus, the computational scheme at the motifs level can provide robustness to the cortical computation.

## Prediction of anatomical constraints on the duration of WM

Elston and his collaborators conducted a series of investigations to measure the spine count of basal dendrites of layer 2/3 pyramidal neurons in macaque [22,23]. This spine count represents one of the sources of microcircuit differences between early sensory cortex and associative cortex [4]. Layer 2/3 pyramidal cells in PFC have as many as 16 times more spines than in V1, as a result, the highly spinous cells in PFC may integrate many more inputs than cells in sensory areas such as V1, TE, and 7a [22,23]. Spines are protruding structures on dendrites that make contact with the axons of other neurons, forming synaptic connections between neurons that transmit nerve signals. Therefore, the most direct effect of changes in spine count lies in input connectivity (or in-degree) of pyramidal cells. Spine count of basal dendrites in Fig 8A affects connectivity probability between pyramidal cells in the associative cortex. We normalized the

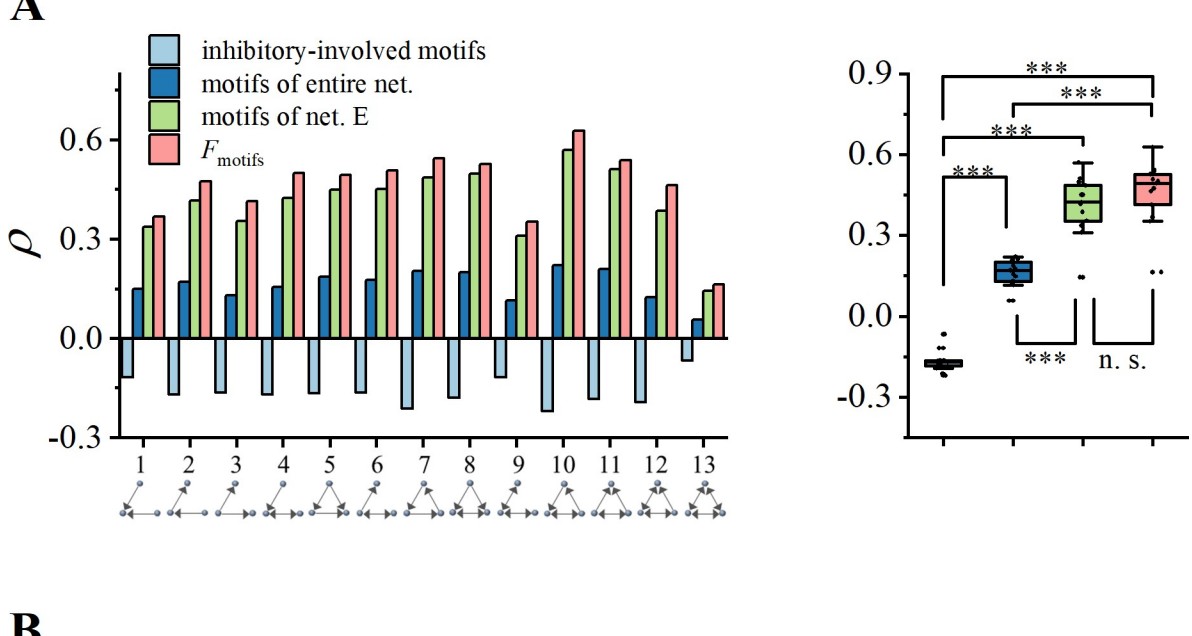

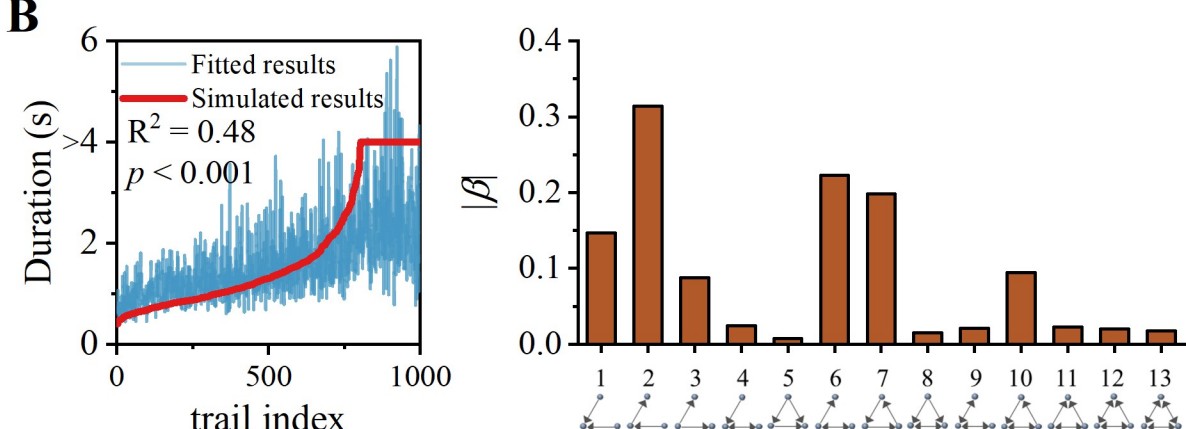

**Fig 7. Regulation of WM-related activity by triadic motifs.** (A) Left panel: Correlation between duration of WM activity and expression of triadic motifs. It is calculated from four perspectives: entire network, excitatory subnetwork, inhibitory involved motifs, and ratio between the latter two. Right panel: The box plots of correlation coefficients from four perspectives demonstrate differences. ***: $p < 0.001$. (B) Left panel: Multivariate regression of triadic motifs ($F_{\text{motifs}}$) on the duration of WM activity. Red line represents our arranged simulated data, and blue line represents corresponding fitted data. Right panel: Regression coefficients $|\beta|$ for the triadic motifs. Motifs involving cycle structure (motifs-7 and 10) and information transmission (motifs-1, 2 and 6) have large regression coefficients.

spine count, resulting in a distribution of connectivity probabilities among pyramidal cells in 24 brain regions, ranging from 15% to 25%. Using the probability distribution, we computed duration of sustained firing activity during delay period, repeating the process 100 times for each brain region.

The distribution of durations across different brain areas is depicted in Fig 8B. Sustained activity is not observed in primary visual pathways (V1, V2, V4, and MT). However, extensive and enduring electrical activity is observed in the prefrontal cortex, which acts as a central hub for cognitive processes (Brodmann Area 46d, 10; 9/46d, 9/46v, 9, 12, 13). Interestingly, this finding is consistent with empirical observations [4]. Recent extensive researches have provided positive and negative evidence for sustained firing encoding in cortical neuron (review

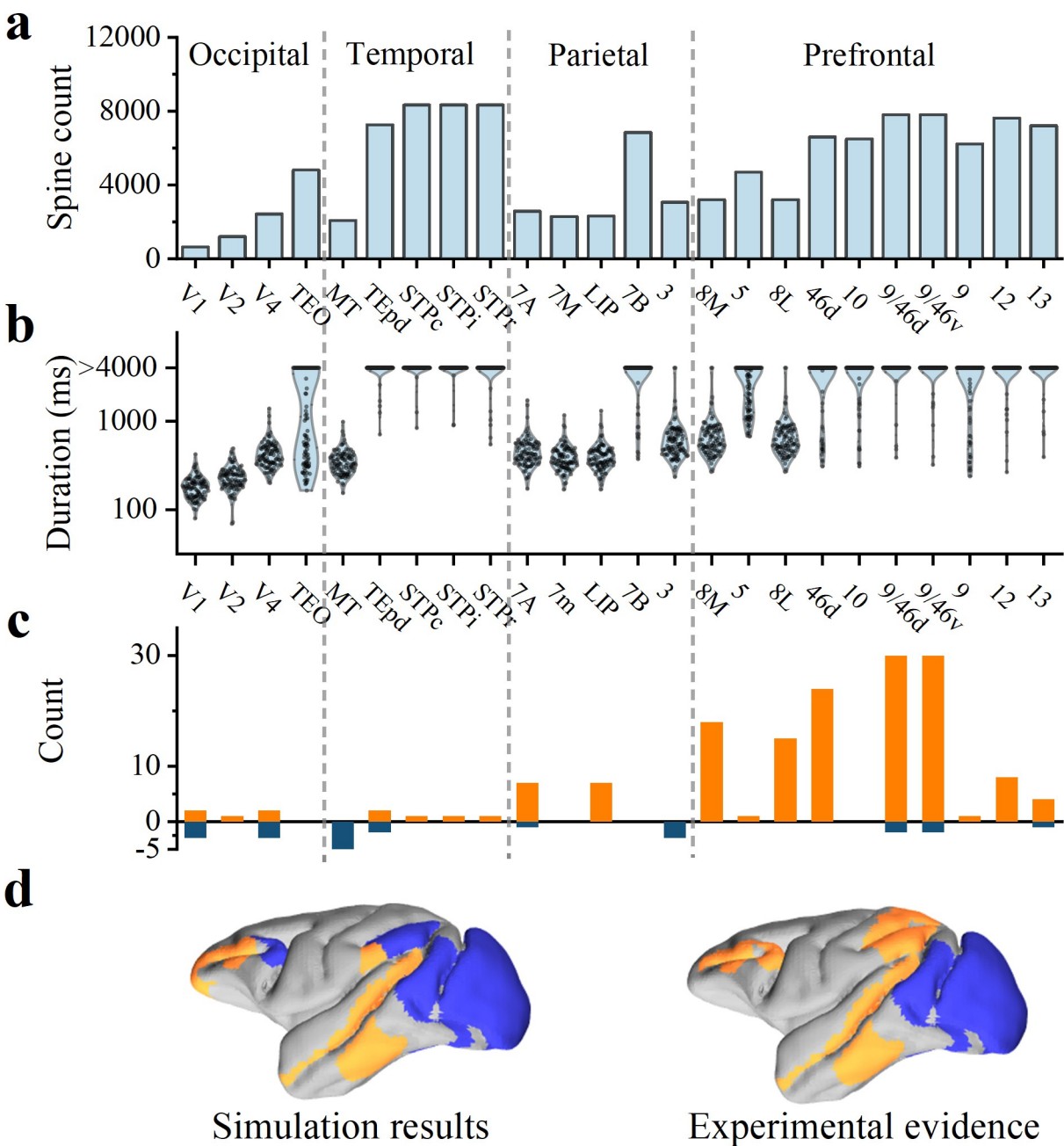

**Fig 8. Anatomical data imposes constraints on cortical model.** (A) Spine count of basal dendrites in 2/3 layer pyramidal neurons of macaque, acquired from Ref. [22,23]. (B) Predictions on duration of WM activity based on synaptic spine. (C) Experimental evidence of WM activity (review in [4]). Orange represents positive evidence, while blue represents negative evidence. (D) Spatial activity map during delayed-period. Left panel displays the results derived from our model, right panel represents the results observed in experiments (adapted from C). Blue and orange areas indicate negative and positive evidence, respectively, regarding the sustained activity of WM. Gray areas indicate unexplored areas within our simulation. All data in Table 1.

in [4], Table 1). The evidence is presented in Fig 8C, with positive reports represented in orange and negative reports indicated in blue.

The presence of seemingly positive evidence in V1, V2, and V4 appears to be attributed to the tasks performed by experimental subjects and recording methods employed [4]. Activity

**Table 1. The spine count data [22,23] from basal dendrites of layer 2/3 pyramidal neurons and evidence [4] of both positive and negative correlations with WM-related activities in delayed period.**

| Abbreviation of area | Spine count | Number of positive evidence | Number of negative evidence | Region |
|---|---|---|---|---|
| V1 | 643 | 2 | 3 | Occipital |
| V2 | 1201 | 1 | | Occipital |
| V4 | 2429 | 2 | 3 | Occipital |
| TEO | 4812 | | | Occipital |
| MT | 2077 | | 5 | Temporal |
| TEpd | 7260 | 2 | 2 | Temporal |
| STPc | 8337 | 1 | | Temporal |
| STPi | 8337 | 1 | | Temporal |
| STPr | 8337 | 1 | | Temporal |
| 7A | 2572 | 7 | 1 | Parietal |
| 7M | 2294 | | | Parietal |
| LIP | 2316 | 7 | | Parietal |
| 7B | 6841 | | | Parietal |
| 3 | 3060 | | 3 | Parietal |
| 8M | 3200 | 18 | | Prefrontal |
| 5 | 4689 | 1 | | Prefrontal |
| 8L | 3200 | 15 | | Prefrontal |
| 46d | 6600 | 24 | | Prefrontal |
| 10 | 6488 | | 1 | Prefrontal |
| 9/46d | 7800 | 30 | 2 | Prefrontal |
| 9/46v | 7800 | 30 | 2 | Prefrontal |
| 9 | 6225 | 1 | | Prefrontal |
| 12 | 7622 | 8 | | Prefrontal |
| 13 | 7205 | 4 | 1 | Prefrontal |

observed in these positive reports [77–80] significantly attenuates after cue offset and decays to baseline within 500 ms, which is in contrast to the activity observed in PFC, parietal lobe, and inferior temporal cortex. The latter persist throughout delay period. Our results also suggest that spiking activity of V1, V2, and V4 can persist for several hundred milliseconds after cue offset (as shown in Fig 8B). Therefore, our simulation results in the occipital lobe align with the experimental results [4,77–80]. Similarly, temporal lobe also demonstrates a significant level of agreement.

In the prefrontal cortex areas 8L and 8M (also known as the frontal eye fields, FEF), our predictions are in complete opposition to experimental observations. We regard that FEF areas require the support from other areas (in form of strong excitation) to maintain delayed activity, rather than depending solely on their local structural properties. Similarly, in parietal lobe, our predictions are also inconsistent with experimental observations. This provides further evidence for distributed WM [21]. Furthermore, our results also provides a testable prediction: electrophysiological recordings in macaques can be used to test whether the FEF and parietal areas indeed require strong excitatory input from the frontal lobe to sustain WM-related activity.

## Discussion

In conclusion, we investigated the relationship between the maintenance of delay-period activity and the local network structure in a WM task. We constructed a cortical neural model

replicating WM features. During delay period, sensory cortex is silent, while associative cortex displays persistent firing. Transitions between silent and persistent activity depend on neurotransmitter gradients, correlating with the WM network structure gradient [38,39].

Our investigation revealed bistability during the transition from silent activity to persistent activity, influenced by neural network topological structure. The duration of WM activity correlates positively with the small-worldness of network structures, which is consistent with the observed small-world organization of brain networks across multiple scales [27–30]. Analysis of excitatory-inhibitory balance in the associated cortex reveals the modulation of delayed-period activities through interaction of recurrent excitation, disinhibition, and lateral inhibition. Multiple regression analysis indicates that small-worldness, excitatory-inhibitory balance, and cycle structures play a more significant role in sustaining WM-related activity, while rich-hub has a relatively weaker impact. Similar findings were also observed in the analysis of triadic motifs in the network. The overexpression of motifs representing information transmission and cycle structures can aid in sustaining WM activity.

Using anatomical data [22,23] and our model, we predicted the duration of WM activity across brain regions. The results obtained from occipital lobe, temporal lobe, and select regions of prefrontal cortex were consistent with experimental observations [4]. The results from parietal lobe and FEF (Frontal Eye Fields) region, however, were in conflict with experimental findings. It means that parietal lobe and FEF rely on strong excitatory support from other brain regions. Therefore, maintenance of WM activity relies on the interaction of local networks and distributed networks [21]. Furthermore, from a perspective of brain structure, brain wiring is predominantly local, with interregional connections accounting for only 20% of total connections [81]. Longer axonal projections incur higher material and energy costs [82–84]. The extensive long-range connections between brain regions contradict the principles of efficient wiring in the brain [24]. Therefore, it further underscores the need to balance WM function through both global and local mechanisms.

Our results highlight the dependence of delay-period activity on local network structure in WM tasks, which has been overlooked in previous studies. Furthermore, by adding rules affecting local structure, such as synaptic plasticity [10,85], observing this structural and functional evolution can further understand the underlying mechanisms [86]. The concept that delayed-period activity maintains WM function is gradually being questioned. Delay-period activity in the PFC is not always essential for maintaining WM, and it can be re-established when attention is refocused on task-relevant content [87]. Therefore, WM does not seem to rely on delay-period activity, and is maintained in the form of "activity-silence". Lundqvist et al. demonstrated that WM exhibits discrete oscillatory dynamics and spiking, rather than sustained activity [88]. In addition, they propose a new concept of spatial computation in which beta and gamma interactions lead to delayed-period activity flowing spatially [89]. We do not emphasize that the maintenance of delay-period activity is equivalent to the WM function, but it intuitively depends on the local network structure. In addition, the WM function involves multiple processes. How the local network structure regulates specific WM functions, such as multi-task collaboration, task distractor, and dynamically update memory content, needs to be further explored in the future.

In previous studies on large-scale network modeling of primate cortex [20,21], the collective behavior of neural populations was derived from an equation describing the evolution of firing rates, which significantly simplifies computational complexity but loses topological characteristics of local circuits. The connections between neurons are influenced by type-specific [32] and morphologies [33] of neurons. Therefore, brain regions with different functions may exhibit distinct topological circuits. Similar to the recently extensively studied neurotransmitter gradients [38,39], in the future, with advancements in experimental techniques, topological

circuit gradients across the brain should also be further measured. It is crucial for further elucidating the relationship between cognitive functions and network structure.

Therefore, by integrating connectomic data and models at the microscale, we can reveal organizational principles of neuronal networks and understand the relationship between anatomical structure and computational function [24]. Furthermore, the complexity of nervous system presents new challenges to the field of network science. To accurately constrain network dynamics, it is necessary to consider physiological characteristics of neurons and synapses. Different types of neurons represent distinct processes, and the connections between nodes and synapses can undergo dynamic changes (synaptic plasticity) [10]. These complexities require the development of novel network metrics.

## Supporting information

**S1 Fig. Deterministic relationship between the kinetic properties of neurons and intensity of cue.** The firing thresholds of pyramidal cells and interneurons are both set at 0.7 nA. When a cue with an intensity of 2 nA is applied to pyramidal cells, neurons exhibit spiking state at a frequency of 52.7 Hz.
(TIF)

**S2 Fig. Evidence of E/I ratio modulation on duration of WM activity at the single-neuron scale.** In 190 networks capable of sustaining electrical activity throughout the entire delay period, the duration of individual neuron's electrical activity is correlated with the E/I ratio. The duration of neuron's electrical activity exhibits a bimodal distribution (the right-side distribution subplot). Neurons with a high E/I ratio tend to sustain electrical activity, while those with a low E/I ratio terminate it rapidly. However, the generation of synaptic currents requires activation of presynaptic neurons. Thus, there exists a range of E/I ratio parameters where both silent and spiking neurons coexist. This suggests that beyond examining interactions between neurons pairwise, higher-order network interactions need to be considered.
(TIF)

**S3 Fig. The correlation matrix of the motifs and E/I ratio.** The size of the dots represents the correlation between variables. There is a high collinearity among the motifs in A) and B), while there is no collinearity among the in-degrees of neurons in C).
(TIF)

**S4 Fig. Small-worldness of the network regulates the duration of WM.** Dependence of duration of WM activity on A) clustering coefficient $\langle Cc \rangle$ and B) shortest path length $\langle l \rangle$ of network is explored. Topological properties of cortical networks are examined from three perspectives: entire network, excitatory subnetwork, and ratio $R$ between the two. It is important to note that the choice of measured variables significantly impacts the correlation observed between these variables and duration of WM activity. C) Spearman correlation between small-worldness and duration of WM activity under three perspectives ($p < 0.001$ for all). Upon examining small-worldness in the entire network, we found that duration is almost uncorrelated with both Cc and SPL. Additionally, probability distributions of the two states exhibit a significant overlap (left panel). However, in excitatory subnetwork, results show a correlation, and the probability distributions of two states separate (middle panel). The relationship between the ratio of two networks (excitatory subnetwork and entire network) and the duration of WM activity reveals a stronger correlation (right panel), it can also be further validated in C), where duration is positively correlated with Cc and negatively correlated with SPL.
(TIF)

**S5 Fig. The mean and standard deviation of expression levels for 13 motifs in 1000 networks.**
(TIF)

## Author Contributions

**Conceptualization:** Dong Yu, Ziying Fu, Ya Jia.

**Formal analysis:** Dong Yu, Tianyu Li, Xuan Zhan, Lijian Yang.

**Funding acquisition:** Ya Jia.

**Investigation:** Dong Yu, Tianyu Li, Ziying Fu.

**Software:** Qianming Ding, Yong Wu.

**Supervision:** Ya Jia.

**Validation:** Qianming Ding, Yong Wu.

**Visualization:** Xuan Zhan, Lijian Yang.

**Writing – original draft:** Dong Yu.

**Writing – review & editing:** Ya Jia.

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
