## [Decision Letter · Decision Letter 0]

10 Jul 2024

Dear Prof. Jia,

Thank you very much for submitting your manuscript "Working memory function is determined by synapses and neural network topology" for consideration at PLOS Computational Biology. As with all papers reviewed by the journal, your manuscript was reviewed by members of the editorial board and by several independent reviewers. The reviewers appreciated the attention to an important topic. Based on the reviews, we are likely to accept this manuscript for publication, providing that you modify the manuscript according to the review recommendations.

Your manuscript was reviewed by two reviewers. Both are positive about the work however they have raised some concerns which require a revision of the text to clarify some terminology, relate the model to biology of the brain networks and better align the work with the recent literature. Moreover, it is important that you share the code of the simulations. For more details please refer to the comments from the two reviewers. We look forward to reading your revision.

Sincerely,

Arvind Kumar, Ph.D.

Academic Editor

PLOS Computational Biology

Daniele Marinazzo

Section Editor

PLOS Computational Biology

Your manuscript was reviewed by two reviewers. Both are positive about the work however they have raised some concerns which require a revision of the text to clarify some terminology, relate the model to biology of the brain networks and better align the work with the recent literature. Moreover, it is important that you share the code of the simulations. For more details please refer to the comments from the two reviewers. We look forward to reading your revision.

Reviewer's Responses to Questions

**Comments to the Authors:**

Reviewer #1: Thank you! This was a nice read, related to my own work, so it was a pleasure. This paper should get published. Its a nice analysis how the (bi)stability of a biophysically detailed network is affected by by overall recurrent network connectivity, connection motifs, cycles, and receptor distributions. Some of these link to known cortical neuroscience, which makes this paper relevant to the field. The model description is reasonably clear, and the results presented in a straight-forward manner. However, I have some critical things to say that - in my mind -demand a few changes to the introduction and the discussion sections in particular, perhaps even the title.

Regarding the introduction

It is initially quite unclear what they mean by “synaptic gradients”, which could mean a lot of different things. Similarly, “aligning with neurotransmitter gradients within the WM network”, is quite unclear. These terms are not nearly as common as the authors perhaps assume, and deserve some concise two sentences of clarification to avoid reader confusion. For example, the later is not really a gradient of the neurotransmitter – like a chemical concentration gradient - , but much rather an experimentally observed gradient in the expression of specific transmitter receptors across the cortical hierarchy. The author summary, discussion and references eventually make this clear, but an introduction should not need an “introduction to the introduction”, but should clarify the scientific context of the study in unambiguous terms.

Topology also has a different meaning to researchers of the sensory cortical hierarchy, as topological connections preserve the nearest-neighbor relationships in the receptive fields across a sensory map across cortical areas. The authors instead use the term as synonymous with “structure” and based on graph-theoretical analysis of the recurrent connectivity within the local network, rather than across areas.

Regarding the Results:

The fact that activity becomes more persistent with small world topology, higher density of recurrent connectivity and its excitatory receptors AMPA and NMDA is qualitatively obvious. So is there a quantitative match with biology, or quantitative prediction in the conductance ranges they show? If so, this would enable further comparisons across models and electrophys data, but it seems that none of these are were biologically constrained, and the reader is left questioning the biological plausibility of the model details. What about the AMPA/NMDA charge ratio in biological cortical neurons(Myme et al. 2003; Rotaru et al. 2011), which would generally suggests much a much weaker NMDA conductance amplitude (NMDAR traces are of course much longer - 50x in this model - , delivering charge over time, rather than high amplitude like AMPA)? Is the model not at odds with the electro-physiological record here?

The analysis of hub-neurons and triadic motives is graph-theoretically interesting and has merit in so far as it links directly to measurements of such quantities in the experimental record. The authors mention that record, and the model advances this discussion largely by providing a proof of concept for the role of these motifs in furthering recurrent activity. However, the authors do not attempt to match biology in any particular way here either, so this part remains a mere correlational analysis with little predictive or explanatory value beyond finding some biological analogies or correspondences.

They model nicely predicts the different areas capacity for persistent activity, based on the expected recurrent connection density, yet given that the model network not clearly constrained otherwise, and the outcome mostly a threshold function on the spine count of basal dendrites in 2/3 layer pyramidal neurons of macaque, it is unclear wether the model is much more than a parameter tuning exercise.

Also: what about re-entrant effect across areas, i.e. the role of inter-area feedback, which may lead to reentrant dynamics? The methods mention that the model includes slightly abstracted interregional connections, but does not specify how exactly these connections are drawn, and what effects they have, which highly related models have found to be important for WM encoding and activity stability (Fiebig et al. 2018).

By normalizing biological spine counts the authors condition their connectivity merely on ratios of the quantitative experimental record, despite the fact that HH-neurons would allow for incorporation of a lot more fine-scale electrophysiological detail. Why use detailed biophysical neurons, if the goal is a mostly graph-theoretical analysis of node relationships?

Regarding the Discussion and the Title, I see a paradigm-sized elephant in the room:

The idea that working memory function is primarily about persistent delay activity, and that such activity is synonymous with retention, is an increasingly questioned paradigm. Modern studies going far beyond the original ‘70s findings of persistent activity in macaque PFC during single-item delay maintenance, have since questioned how persistent any observed activity increases really are, if evidence for it are not mostly artifacts of large binning, intra- and inter-trial averaging, and reanalysis of old data has since shown that any increases in cell firing rates are generally much smaller (+3 Hz above baseline perhaps) than suggested by most recurrent network models, including this one (+30Hz or more for several seconds across an entire local network, which is unheard of in real recordings) (Shafi et al. 2007; Lundqvist et al. 2018).

Further, work of the last two decades has shown how these “persistent” signatures disappear with multi-item maintenance or distractors, without necessarily impacting eventual recall performance, when item-selective activity returns. This calls into question whether lasting activity is not primarily a signature of attention (or readout), rather than the memory engram itself. Ideas of “activity-silent” working memory(Stokes 2015), that may be revealed by unrelated strong stimuli(Rose et al. 2016), bursty-maintenance(Lundqvist et al. 2016), rely on synaptic memory (Mongillo et al. 2008; Lansner et al. 2023) and even local spatial computing in working memory networks(Lundqvist et al. 2023) have since gained traction in model work as well (Fiebig and Lansner 2017; Manohar et al. 2019).

A very critical reader may well conclude that the title of this paper is somewhat misleading. It is a careful graph-theoretical analysis of bistability in a biophysically detailed recurrent neural network. As such, it has merit, particularly where the model is actually constrained by neuroscience, but the link with working memory, as a term from cognitive science, is tenuous at best. Working memory implies much more than a single-item short term memory buffer, yet the authors do not even attempt to encode or maintain more than one item, apply distractors, dynamically update memory content, or use their “working memory” for any other task that might link tentatively to the executive functions implied by the term working memory. Given that the authors also uphold a now questionable paradigm about the centrality of persistent activity for WM maintenance, they should at least mention the ongoing controversy in their discussion rather than treat recurrent network (bi)stability as essentially synonymous with working memory.

Reproducibility

The most important aspects to rebuild this model from scratch are well laid out in the Methods, yet the Code-availability link is broken, it seems, which is a pity. I would have loved to take a look to find the answers to some of my questions myself.

Reviewer #2: This is solid work, but poor presentation. I support the publication of this work, even as it is, but I recommend the authors restructure a bit the paper to increase impact. Instead of a long ling of things that impact WM duration, decide what are the main results and focus on that and move the rest to supplementary. Try as much as possible to merge or remove figures and have the strongest message possible on each figure.

Minor:

1. introduce the model equations earlier in the text

2. Previous work focus on much simpler models, which seems good enough to study topological aspects of the connectivity. Justify (and highlight!) the choice for HH.

**Have the authors made all data and (if applicable) computational code underlying the findings in their manuscript fully available?**

Reviewer #1: **No: **seems like the git link to the code is broken

Reviewer #2: **No: **there is a typo in the link

PLOS authors have the option to publish the peer review history of their article (what does this mean?). If published, this will include your full peer review and any attached files.

Reviewer #1: No

Reviewer #2: No

Figure Files:

Data Requirements:

Reproducibility:

References:

---

## [Decision Letter · Decision Letter 1]

14 Aug 2024

Dear Prof. Jia,

We are pleased to inform you that your manuscript 'Maintenance of delay-period activity in working memory task is modulated by local network structure' has been provisionally accepted for publication in PLOS Computational Biology.

Best regards,

Arvind Kumar, Ph.D.

Academic Editor

PLOS Computational Biology

Daniele Marinazzo

Section Editor

PLOS Computational Biology

The two reviewers are happy with the revision. So we can formal accept the manuscript for submission as it is.

Reviewer's Responses to Questions

**Comments to the Authors:**

Reviewer #1: I am very happy with the point by point response the authors gave to my criticism and suggestions, and like the authors, I believe that this has substantially improved the manuscript. Lets get this out now!

Reviewer #2: I support the publication of this manuscript as it is

**Have the authors made all data and (if applicable) computational code underlying the findings in their manuscript fully available?**

Reviewer #1: Yes

Reviewer #2: Yes

PLOS authors have the option to publish the peer review history of their article (what does this mean?). If published, this will include your full peer review and any attached files.

Reviewer #1: **Yes: **Florian Fiebig

Reviewer #2: No

---

## [Editor Report · Acceptance letter]

27 Aug 2024

PCOMPBIOL-D-24-00689R1 

Maintenance of delay-period activity in working memory task is modulated by local network structure

Dear Dr Jia,

I am pleased to inform you that your manuscript has been formally accepted for publication in PLOS Computational Biology. Your manuscript is now with our production department and you will be notified of the publication date in due course.

With kind regards,

Jazmin Toth
